# Creatinine Trends and Patterns in Neonates Undergoing Whole Body Hypothermia: A Systematic Review

**DOI:** 10.3390/children8060475

**Published:** 2021-06-04

**Authors:** Noor Borloo, Anne Smits, Liesbeth Thewissen, Pieter Annaert, Karel Allegaert

**Affiliations:** 1Department of Development and Regeneration, KU Leuven, Herestraat 49, 3000 Leuven, Belgium; noor.borloo@student.kuleuven.be (N.B.); anne.smits@uzleuven.be (A.S.); 2Neonatal Intensive Care Unit, UZ Leuven, Herestraat 49, 3000 Leuven, Belgium; liesbeth.thewissen@uzleuven.be; 3Department of Pharmaceutical and Pharmacological Sciences, KU Leuven, Herestraat 49, 3000 Leuven, Belgium; pieter.annaert@kuleuven.be; 4Department of Clinical Pharmacy, Erasmus MC, Postbus 2040, 3000 GA Rotterdam, The Netherlands

**Keywords:** creatinine, cystatin C, asphyxia, whole body hypothermia, acute kidney injury, renal clearance, kidney function

## Abstract

Many neonates undergoing whole body hypothermia (WBH) following moderate to severe perinatal asphyxia may also suffer from renal impairment. While recent data suggest WBH-related reno-protection, differences in serum creatinine (Scr) patterns to reference patterns were not yet reported. We therefore aimed to document Scr trends and patterns in asphyxiated neonates undergoing WBH and compared these to centiles from a reference Scr data set of non-asphyxiated (near)term neonates. Using a systematic review strategy, reports on Scr trends (mean ± SD, median or interquartile range) were collected (day 1–7) in WBH cohorts and compared to centiles of an earlier reported reference cohort of non-asphyxia cases. Based on 13 papers on asphyxia + WBH cases, a pattern of postnatal Scr trends in asphyxia + WBH cases was constructed. Compared to the reference 50th centile Scr values, mean or median Scr values at birth and up to 48 h were higher in asphyxia + WBH cases with a subsequent uncertain declining trend towards, at best, high or high–normal creatinine values afterwards. Such patterns are valuable for anticipating average changes in renal drug clearance but do not yet cover the relevant inter-patient variability observed in WBH cases, as this needs pooling of individual Screa profiles, preferably beyond the first week of life.

## 1. Introduction: Perinatal Asphyxia, Whole Body Hypothermia, and Renal Impairment

Perinatal asphyxia is a condition at delivery, driven by deprivation of oxygen sufficiently long enough to potentially result in several sequelae-like hypoxic-ischemic encephalopathy (HIE) or cerebral palsy [1,2]. Perinatal asphyxia is the final result of different types of events such as placental blood flow disruption, prolonged labor, or compression of the umbilical cord. All of these events result in reduced circulating blood oxygen, while asphyxia is the most common cause of encephalopathy in neonates. Hypoxic-ischemic encephalopathy is hereby characterized by both clinical- and biomarker-based (laboratory, electro-encephalography (EEG)) evidence of acute or subacute brain injury (encephalopathy) due to the fact of intrapartum or late antepartum brain hypoxia and ischemia [2,3]. It still accounts for a relevant proportion of neonatal deaths, especially when we focus on causes of mortality in (near)term neonates [4]. Whole body hypothermia (WBH) is an effective intervention to reduce mortality, even more pronounced in low-income countries [5,6]. Approximately 0.5% of live born neonates are affected by perinatal asphyxia, while 0.18% are diagnosed with HIE, either mild, moderate, or severe [2,7]. Both the hypoxic-ischemic phase as well as the reoxygenation–reperfusion event of perinatal asphyxia contribute to the presence and extent of brain injury [8,9]. The inflammatory response is another relevant contributor to neurodevelopmental outcomes in newborns following WBH [10].

It is well known that asphyxia is a multi-organ disease, as asphyxia also affects multiple other organs besides the central nervous system. This is because the initial cardiovascular response to oxygen depletion results in organ-specific vasoconstriction. This subsequently results in the redistribution of oxygen from non-vital organs to the heart and brain, while other organs, such as the intestines, liver, or kidneys, are underperfused [11,12,13]. As vasoreactive organs, the kidneys are very sensitive to oxygen deprivation, and this commonly (39–42%) results in oliguric or non-oliguric acute kidney injury (AKI) in this setting [14].

Acute kidney injury was observed in approximately 1.5% of admissions in a single neonatal intensive care unit. When these authors focused on the subgroup of term neonates, perinatal asphyxia was the most common cause (72% of 72 term cases) of AKI, followed by congenital kidney and urinary tract malformations (CAKUT, 8.3%), congenital heart disease (6.9%) or sepsis (2.8%) [15]. This confirms to a large extent the patterns on incidence and risk factors of early onset neonatal AKI as described in the AWAKEN (Assessment of Worldwide Acute Kidney Epidemiology in Neonates) study [14,16]. To further illustrate the interrelatedness of this multi-organ disease, Cavallin et al. recently reported on the prognostic role of AKI on long-term outcome in HIE infants. The presence of AKI hereby served as a reliable indicator of death or long-term disability in HIE neonates undergoing WBH, while its absence was not a guarantee for a favorable long-term outcome. In essence, this suggests that this is a specific but not very sensitive marker [17]. Along the same line, a higher positive fluid balance (oliguria, the other indicator of AKI, besides serum creatinine (Scr) and Scr trends) was associated with death or moderate to severe brain injury in cases following WBH because of HIE [18].

Whole body hypothermia has become the standard neuroprotective treatment for moderate to severe perinatal asphyxia since 2010, supported by meta-analytical evidence that WBH significantly reduces both mortality and morbidity (a relative risk (RR) reduction of 25% for survival with normal neurocognitive outcome, equal to a number needed to treat (NNT) for an additional beneficial outcome of seven) in moderate to severe HIE cases [19,20]. Interestingly and in addition to improved neurodevelopmental outcome following WBH, there is recent meta-analytical evidence that WBH also reduces the incidence of AKI ((RR) 0.81, 95% confidence interval (CI) 0.67–0.98, NNT = 7). Consequently, besides improving mortality and neurocognitive outcome, WBH also seems reno-protective if we focus on the short-term outcome (i.e., AKI), while data on long-term renal outcome are not yet reported [21]. Consequently, there is clinical relevance in describing the overall Scr trends as reported in cohorts undergoing WBH to better understand the postnatal Scr trends and to compare these patterns with the centiles of a reference data set in non-asphyxiatic (near)term neonates. This should also enable clinicians to plot or compare observations in individual patients to average trends. Furthermore, such findings can inform us to guide pharmacotherapy or to further elaborate the link between Scr trend observations and outcome.

Different studies have quantified the impact of asphyxia and WBH on the elimination of drugs that are dependent on the glomerular filtration rate (GFR). Renal elimination was reduced (from −25% up to −40%), as reflected by the reduced gentamicin clearance by 25–35.3% with a progressive trend to normalization after termination of WBH (>72 h after initiation of WBH). A similar pattern of decreased clearance (−40%) has been described for amikacin during WBH, or even up to −60% when these estimates were based on mannitol clearance data [22,23,24,25,26].

Although there are different and perhaps more performant biomarkers to quantify renal function like cystatin C, Scr is still the most frequently used and accessible biomarker to evaluate renal (dis)function or GFR including in the neonatal intensive care setting [27,28,29,30]. Assuming stable creatinine synthesis (reflecting muscular mass), an increase in Scr reflects poorer GFR. Despite its common use, this biomarker has some disadvantages.

At birth, creatinine still somewhat—but not fully—reflects maternal Scr levels [31]. This is because creatinine at birth is somewhat higher in the most mature neonates compared to the preterm cases with a positive correlation with birth weight or gestational age, despites stable maternal Scr values [31]. Furthermore, there is no active renal tubular secretion in neonates yet but rather passive renal tubular back leak, so that creatinine clearance does not yet entirely reflect GFR. This is in contrast to older children or adults, where active tubular creatinine secretion results in minor overestimation of the “true” GFR when based on creatinine clearance [27,28,29,30]. Another specific issue is the variability in Scr assays, as the Jaffé—and to a certain extent—the enzymatic assays can be affected by the plasma matrix in neonates, with a lower albumin concentration, and a commonly higher bilirubin [27,28,29,30]. Validation of those assays to isotope dilution mass spectrometry (IDMS) should reduce this assay-related variability [32].

Despite these limitations, Scr values to estimate GFR are, at present, routinely used worldwide in neonatal clinical care; thus, we decided to focus our systematic review on Scr trends in WBH cohorts but also search for cystatin C values. As recent data suggest a WBH-related reno-protective effect, we aimed to document Scr trends in asphyxiated neonates undergoing WBH based on a systematic search strategy on published cohorts and compared these values to the centiles of a reference Scr data set restricted to (near)term neonates (≥36 weeks gestational age) of a similar gestational age [21,27].

## 2. Materials and Methods

This systematic review has been performed using the Preferred Reporting Items for Systematic Reviews and Meta-Analysis (PRISMA) guidelines [33]. The literature search was performed on the first of May 2020 using Embase, PubMed, Cochrane Libraries, and Web of Science as information sources. The search was limited to the English language, and papers had to report on values or trends in Scr or cystatin C in human newborns with asphyxia and undergoing WBH. An overview of the manually performed search strategies is provided in Figure 1. If the article was retained based on the title and abstract, and in the event that the full text was not found (Open Access, University Libraries Leuven, Rotterdam, or their network), we contacted the corresponding author to request a full-text version.

A screening, eligibility, and inclusion assessment were performed independently by two reviewers (NB, KA) with subsequent discussion in the event of a disagreement until a consensus was reached. Inclusion was based on the (1) population: asphyxiated newborns; (2) intervention: undergoing or underwent whole body therapeutic hypothermia (irrespective of add-on interventions such as additional selective head cooling); (3) outcome: Scr or cystatin C values reported in the first week of life. Only articles written in English and with access to the full text were included. Postmortem and animal studies, congress abstracts without full-text availability, and neonates with a history of cardiac problems or surgery were not included, in line with the other reported risk groups for AKI in (near)term neonates [15,16].

Data items extracted were journal, type of study (observational, interventional), year of publication, duration of hypothermia, population (demographic data, number), intervention (hypothermia, either WBH or selective head cooling (SHC), as only data in WBH cases were retained), control and outcome (Scr, cystatin C), and the equipment used to induce hypothermia or for the Scr or cystatin C measurements the assays involved. The demographic data extracted were gestational age, birth weight, male or female, and APGAR score at 5 and 10 min of age. The individual papers were also screened for data on the Thompson score and lactate measurements.

The aim was to extract Scr data, as reported, per individual day in either mean ± SD or median and range in the first week of postnatal life (day 1–7) in asphyxia + WBH cases (irrespective of the presence of other interventions such as additional selective head cooling). When data were not reported in the source paper, corresponding authors were contacted and asked if they were willing to provide the needed information. In the absence of a reply from the corresponding author, we used WebPlotDigitizer (by Ankit Rohatgi), to extract data from the charts as provided [34]. Alternatively, when the population of interest (i.e., asphyxia + WBH) was reported in subgroups with a number of cases and a mean or median (such as AKI versus non-AKI cases), we calculated the proportional mean/median and standard deviation for the full population of interest based on the observations reported in the subgroups. The Scr values per day are reported uniformly in mg/dL, following conversion from µmol/L, by dividing by 88.4, when appropriate. Furthermore, we used the Strengthening the Reporting of Observational Studies in Epidemiology (STROBE) statement checklist to assess the quality of the included studies [35].

We subsequently compared our obtained results with the centiles and Scr trends from an earlier reported data set [27]. To ensure similarity in clinical characteristics besides the asphyxia + WBH, we restricted this data set to 1456 Scr observations in a reference cohort of (near)term neonates (≥36 weeks gestational age) without asphyxia (UZ Leuven) in the first week of postnatal life (Table 1) [27].

Finally, as an additional “external” effort to find reports to further refine the Scr pattern described based on the cohorts retained in the meta-analysis, we conducted another non-systematic search for cohorts published after finalization of the search (1 May 2020).

## 3. Results

Our search strategy identified 213 articles: 41 articles with the PubMed search, 126 hits as result of the Embase search, nine reviews and 18 trials from the Cochrane search, and 18 results via Web of Science. Among these, 35 articles were duplicates. After the application of inclusion and exclusion criteria, 43 articles were eligible by title and abstract. Decisions to exclude papers on eligibility were generally based on the earlier mentioned in- and exclusion criteria (such as postmortem pathology or animal experimental data) or language. One conference abstract (of four retained during eligibility, and none after screening) was not retrieved in full. Of these 43 articles, 13 articles were retained after screening the full-text version based on the approach earlier described. Exclusion reasons during this screening procedure, including the search strategy and its outcome in the consecutive steps of the process, are provided in Figure 2 as a PRISMA flow diagram.

We hereby included one paper that was not part of the results of the initial search strategy but was suggested by one of the corresponding authors of the eligible studies when we contacted her for a full-text version (P. S. Wintermark) of her paper.

The characteristics of all included articles are listed in Table 2 [36,37,38,39,40,41,42,43,44,45,46,47,48]. These articles included data on WBH neonates diagnosed with HIE, admitted to a neonatal intensive care unit within 6 h after birth. Whole body hypothermia for 72 h was used in all studies, while, in some, selective head cooling was also applied in addition to WBH, as this was a RCT comparing selective head cooling, either in addition to WBH or not. Based on the inclusion criteria, only cases exposed to WBH were retained for analysis [48]. Patient inclusion criteria for all studies were infants ≥36 weeks of gestation (except for Gupta et al., >35 weeks + birth weight ≥ 1800 g) [42], with a diagnosis of moderate to severe perinatal asphyxia within 6 h after birth. The HIE diagnostic criteria are also listed per study in Table 2; they combine both clinical and EEG criteria.

Patient characteristics per study are listed in Table 3. Thompson score and lactate measurements reflecting disease severity were not available, so they were not retained in the reporting. Two out of the 13 included studies [39,47] used the HIE inclusion criteria from the Total Body Hypothermia (TOBY) trial, and three out of the 13 studies [37,38,44] used the inclusion criteria of the Cool Cap trial [49,50,51]. The clinical inclusion criteria of the Cool Cap protocol were similar to the TOBY protocol, be it with minor differences for the EEG inclusion criteria. The TOBY trial included neonates with signs of at least 30 min of amplitude integrated EEG recording that showed abnormal background (amplitudo) aEEG activity or seizures, while the Cool Cap trial included neonates with signs of at least a 20 min duration of amplitude integrated EEG with cerebral function monitoring (aEEG/CFM) recording that showed abnormal background aEEG/CFM activity or seizures [49,50,51].

Eleven out of the 13 included studies used WBH as a single cooling strategy, while the two other papers related to the Cool Cap trialadded (randomized) selective head cooling to WBH. In all studies, WBH was initiated within 6 h after birth and core temperature was kept at 33–34 °C for 72 h. After cooling, rewarming was set at 0.5 °C per hour to reach normal body temperature within 6–8 h. Seven articles followed the National Institute of Child Health and Human Development (NICHD) criteria for WBH [41]. The other five studies followed a similar protocol (Table 2 provides additional details for these studies) [36,37,38,39,40,41,42,43,44,45,46,47,48]. Battin et al. used selective head cooling as a cooling mechanism, combined with mild systemic hypothermia [48]. This was reached using different cooling caps, precooled to 10 °C. Rectal, fontanelle, and nasopharyngeal temperatures were continuously monitored, keeping rectal temperatures for 72 h at 34.5–35.5 °C or normothermia, dependent of the study group allocation. Like in the NICHD protocol, rewarming was performed with 0.5 °C per hour, eventually reaching normal body temperature.

Finally, related to the STROBE quality assessment, all included studies performed well on this checklist. Potential bias was mentioned in two cohorts [36,40]. Limitations were explicitly discussed in 10 out of the 13 studies.

We listed the available Scr data—including information related to the variability, such as the standard deviation, interquartile range or range—per day (Table 4, data listed in mean ± SD or median + interquartile or range). We hereby would also like to mention that data on cystatin C values or trends in perinatal asphyxia + WBH were not retrieved in this meta-analysis, so they cannot be provided.

In both data sets, we noticed a similar pattern with an initial increase from birth onwards over the first day of postnatal life and a subsequent decline after the first day of postnatal life, confirming the patterns earlier described in other cohorts of (near)term neonates [52,53,54].

When comparing both data sets by visual inspection (Figure 3), we noticed that all the mean/median Scr data points (*n* = 18) values at birth and the first two days (until 48 h) were above the 50th centile values of the reference data set for the same time points. Already before initiation of WBH, the Scr values were higher (between 0.8 and 1.2 mg/dL), with a further rising trend to the first day of postnatal life, with Scr values equivalent to the 90th centile and beyond when compared to the reference data set (Figure 3, Table 4).

We subsequently noticed a decline in both data sets with many more uncertainties concerning differences in values and trends. However, in some WBH cohorts, the mean or median values remained above the 90th centile in the second part of the first week of postnatal life, thus extended beyond WBH finalization, with Scr values between 0.4 and 0.8 mg/dL (with some cohorts crossing the centiles to the 50th–90th centile range).

As the result of an additional “external” effort to find more recently published (after 01 May 2020) reports to further refine the Scr pattern described based on the cohorts retained in the meta-analysis, Mok et al. (published September 2020) published another retrospective observational cohort that—post hoc—also meets all inclusion criteria of our literature review [55]. As an “external validation”, we plotted these Scr observations (day 1–7) in Figure 4 on top of the data initially summarized in Figure 3. This more recently published data set further adds uncertainty to the pattern in either the values or trends for Scr in the second part of the first week of postnatal life.

## 4. Discussion

In addition to the fact that kidney function is a prognostic factor for survival and neurocognitive outcome in neonates born with asphyxia, Scr trends are also relevant for adjusting pharmacotherapy and fluid management to the individual neonatal kidney function [17,18]. So far, there is no information on Scr trends for neonates with perinatal asphyxia + WBH, while a recent meta-analysis provided evidence for a specific reno-protective effect of WBH (NNT 7 to prevent AKI in this specific setting) [21].

This current systematic review intended to provide an overview of the dynamic Scr trends for (near)term neonates born with asphyxia and treated with WBH within the first 6 h after birth. We hereby focused on Scr results described from birth (so before initiation of WBH) until day seven after birth. This time interval is generally classified as “early neonatal life” and covers both the full duration of WBH (72 h), followed by rewarming to reach normothermia. We subsequently compared these results in WBH cases to the daily creatinine values and patterns from non-asphyxia cases, obtained from an earlier published data set [44].

Based on this approach, we noticed that the initial Scr values were already increased before WBH. This is relevant, as the most currently used definitions for AKI and their staging in neonates—such as the Kidney Disease Improving Global Outcome (KDIGO) definition—are also based on proportional Scr increase (stage 1: + 0.3 mg/dL, or 1.5–1.9-fold increase in Scr; stage 2: 2–2.9-fold increase in Scr) from a baseline observation [56]. Based on the current analysis, the first Scr (<6 h, reflecting the baseline value) was already higher when compared to the reference centiles (Table 1). This implies that, technically, a proportional increase needs a higher absolute increase in Scr value in this population to classify for stage 1 or stage 2 AKI [56]. This reflects somewhat of a limitation of the AKI definition. The development of centile values for Scr should be further considered to better capture the maturational versus non-maturational trends in Scr in neonates, but this will necessitate individual observations in a sufficiently large data set [54]. Based on the current uncertainties in the second part of the first week of postnatal life, it is hereby useful to further extend this period to 10 or 14 days to capture the full pattern, as a first take-home message.

Biomarkers, such as cystatin C, are perceived to be better markers of kidney damage, but these are not yet commonly implemented in this specific clinical neonatal setting [27,54,57]. This resulted in the absence of papers in this systematic review reporting on cystatin C trends in asphyxia + WBH cases; therefore, we only could include studies that reported on Scr values and trends during and over the days following WBH. This is another take-home message from this analysis and a call to clinical researchers to assess the potential add-on value of cystatin C to Scr in this population. However, cystatin C also comes with analytical limitations in neonates [58,59]. Furthermore, the quality of the Scr assay reporting was substandard, with only information on the assay in two of the cohorts retained in the meta-analysis, and only one cohort reported on the IDMS traceability [32]. Furthermore, the link between Scr and GFR as a direct measure of renal function is only poorly validated in early neonatal life. However, the link between AKI and with neurocognitive outcome and mortality in this population has repeatedly been reported [7,11,14,17,18].

Our approach and analysis have its limitations, as well as several strengths. Based on the current systematic search strategy, 13 papers met our inclusion criteria and were retained to extract longitudinal mean or median Scr values. Unfortunately, we were not able to further explore Scr variability within and between these cohorts, as this necessitates access to individual data, similar to meta-analysis of individual patient data to facilitate data aggregation or explore trends as recently illustrated by Hage et al. to study the shifts in clinical characteristics of HIE cases in England, Wales, and Scotland [60].

In general, the pattern of postnatal Scr trends with an initial increase up to 24 h, and a subsequent decrease over the first week of postnatal life is similar in asphyxia + WBH cases when compared to (near)term non-asphyxia cases. However, the Scr values in asphyxia cases before cooling are already higher than those of the reference cohort, with higher Scr in the first 48 h of life, with a subsequent poorly described decrease (Figure 3). Based on the available data, it is difficult to draw conclusions on the variability because of limitations in our analysis, while a statistical analysis beyond a simple description of the visual pattern was not feasible.

First, the pattern can differ because of methodological limitations or variability related to the meta-analysis. Individual Scr values were never available, while assays for obtaining those data were only rarely reported (Table 4). For pragmatic reasons, the extracted data were based on median or mean values as reported in the individual cohorts. As described in the methods, if these data were not reported in the individual paper, we contacted the corresponding author (only one author replied and provided this information). Alternatively, data were extracted from the plots or recalculated, if data were only reported in different subgroups of asphyxia + WBH cases. Consequently, the data obtained in the asphyxia + WBH cohorts were limited to mean or median Scr observations for consecutive days, in contrast to the centile Scr data reference cohort, as we were unable to calculate centiles in the asphyxia + WBH cohorts (Figure 3, Table 4). Along the same line, we were not able to explore potential covariates of further raised Scr, such as covariates of disease severity (such as lactate or Thompson score), as such data could neither be retrieved. Furthermore, eight out of the 13 included papers were observational studies. Although this potentially enhances the risk of selection bias, asphyxia + WBH is a very operational and rather reliable inclusion criterion. Besides these limitations, there were also some strengths in this analysis.

A strength of this literature review was the extensive search conducted in four databases, using a hand search to include the maximum number of relevant results, and an independent analysis by two authors. As an additional “external” reflection on the uncertainties in the pattern as described, we noticed that after our search strategy was finalized, the Mok et al. data were plotted (Figure 4) [55]. This more recently published data set provided information on a more blunted decrease of Scr in the second part of the first week of postnatal life. At least, this additional data set further confirms the fact that in future pooling efforts, postnatal age should be further extended beyond the first week of postnatal life to provide more confidence on the Scr patterns in asphyxia + WBH cases. Finally, this paper further supports other recent efforts to also assess non-neurodevelopmental aspects of the outcome (renal, cardiac) in asphyxia + WBH cases, as well suggests incorporating these aspects into the ongoing efforts to develop a core outcome set specific to neonatal encephalopathy trials [21,61].

## 5. Conclusions

Based on a systematic search strategy, 13 papers were retained to construct a provisional pattern on postnatal Scr trends in asphyxia + WBH cases. Compared to a reference cohort of (near) term non-asphyxia cases, mean or median Scr values at birth and the first two days during WBH remained higher in asphyxia + WBH cases, with a subsequent decline to reach at best high or high normal creatinine values from day 4 onwards, be it that this decline was only poorly described. Finally, it also illustrates the limitations of the Scr thresholds used in even the most recently developed AKI stage definition. Such patterns are valuable to anticipate average changes in renal drug clearance but do not yet cover the relevant inter-patient variability observed in WBH cases, as this needs pooling of individual Screa profiles, preferably beyond the first week of life.

## Figures and Tables

**Figure 1 children-08-00475-f001:**
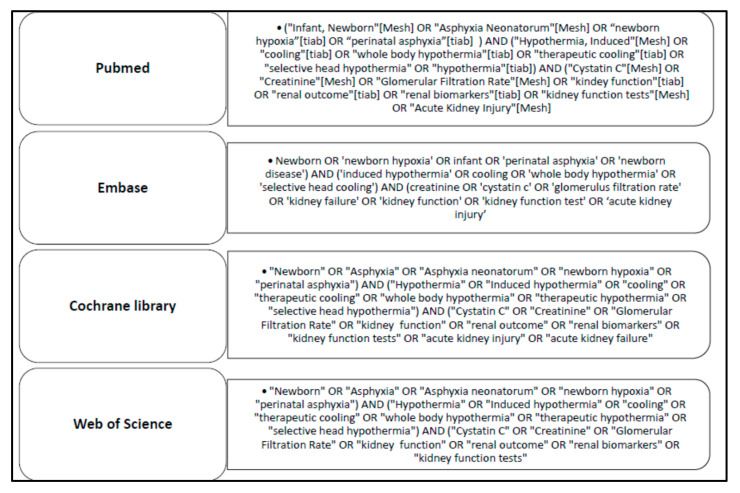
Search strategy as conducted on PubMed, Embase, Cochrane Library, and Web of Science.

**Figure 2 children-08-00475-f002:**
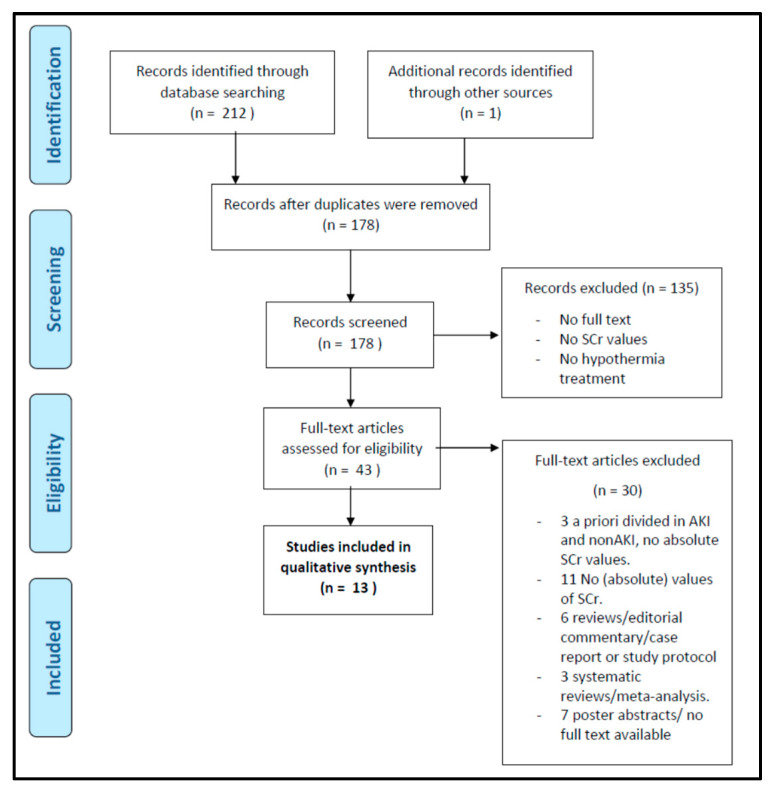
PRISMA flow diagram of the systematic search strategy.

**Figure 3 children-08-00475-f003:**
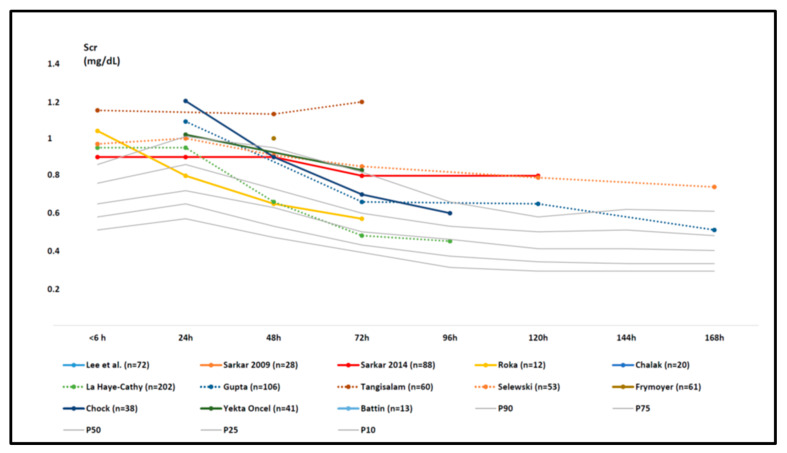
Mean or median serum creatinine (Scr) data as reported in the individual retained studies (colored trends) compared to centile (10th, 25th, 50th, 75th, and 90th centiles) trend values in the reference cohort (Table 1, grey trends) in the first week of postnatal life [27,36,37,38,39,40,41,42,43,44,45,46,47,48]. The study size is provided in the legend (number of cases), and additional characteristics can be retrieved in Table 2 and Table 3. Aspects related to Scr variability (standard deviation or interquartile range) are provided in Table 4.

**Figure 4 children-08-00475-f004:**
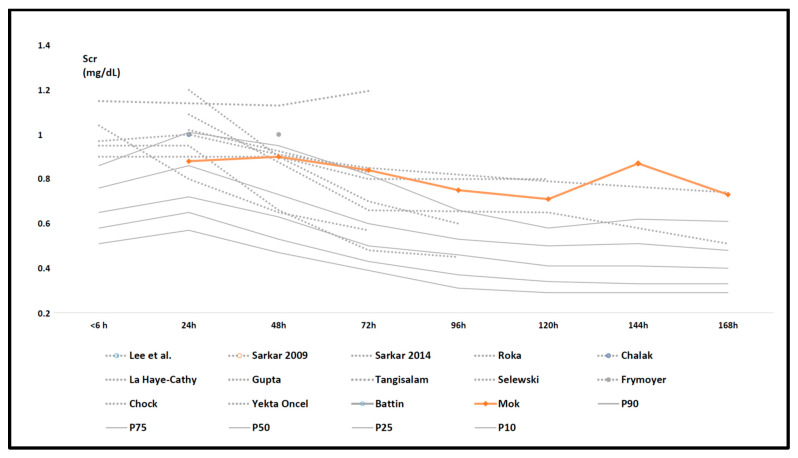
Serum creatinine (Scr) data as reported in the individual retained studies (dashed, grey trends) compared to centile (10th, 25th, 50th, 75th, and 90th centiles) trend values (full grey) in the reference cohort (Table 1) in the first week of postnatal life with the trends in Scr mean values as reported in the Mok cohort (66 cases, orange) added [27,36,37,38,39,40,41,42,43,44,45,46,47,48,55].

**Table 1 children-08-00475-t001:** Centile serum creatinine values (Scr, mg/dL, 90th, 75th, 50th, 25th, and 10th) at birth, and during the consecutive days in early neonatal life (until day 7), based on 1456 Scr observations in 495 (near)term cases (gestational age ≥ 36 weeks), extracted from a previously reported data set of neonates [27].

	<6 h	Day 1	Day 2	Day 3	Day 4	Day 5	Day 6	Day 7
P90	0.86	1.01	0.95	0.82	0.66	0.58	0.62	0.61
P75	0.76	0.86	0.73	0.6	0.53	0.5	0.51	0.48
P50	0.65	0.72	0.63	0.5	0.46	0.41	0.41	0.4
P25	0.58	0.65	0.53	0.43	0.37	0.34	0.33	0.33
P10	0.51	0.57	0.47	0.39	0.31	0.29	0.29	0.29
Number	293	239	215	213	160	129	106	101

**Table 2 children-08-00475-t002:** Study characteristics of the 13 studies retained in the systematic review (HIE: hypoxic-ischemic event; WBH: whole body hypothermia; Scr: serum creatinine; NICHD: National Institute of Child Health and Human Development; SHC: selective head cooling; TOBY: Total Body Hypothermia trial). The group of interest was the asphyxia + hypothermia group, only when based on WBH [36,37,38,39,40,41,42,43,44,45,46,47,48].

Author	n	Study Design	HIE Diagnosis	Hypothermia	Control	Time of Scr Sampling	Additional Information
Lee et al. 2017 [36]	72	Observational, prospective study	(1) Blood gas pH <7.15 or base deficit >10 mmol/L + moderate to severe encephalopathy OR (2) acute perinatal event and 10 min APGAR <5, OR assisted ventilation for 10 min after birth, and moderate to severe encephalopathy	WBH for 72 h (NICHD)	/	Max creatinine measurement between 24 h and 96 h of age	Scr assay: not mentioned Cooling blanket (Mul-T Blanket)
Sarkar et al. 2009 [37]	28	Observational study	Cool Cap trial	WBH for 72 h (NICHD) (*n* = 28)	SHC for 72 h (Cool Cap) (*n* = 31)	24 h, 48 h, 72 h	Scr assay: not mentioned
Sarkar et al. 2014 [38]	88	Retrospective, observational review	Cool Cap and NICHD protocol	WBH for 72 h (NICHD)	/	Baseline, 24 h, 48 h, 72 h, d 5/7	Scr assay: not mentioned
Róka, et al. 2007 [39]	12	Randomized controlled trial	TOBY study	WBH for 72 h (*n* = 12)	Normothermia (*n* = 9)	6 h, 24 h, 48 h, 72 h	Scr assay: not mentioned Cooling mattress—Core temp 33–34 °C
Chalak et al. 2014 [40]	20	Prospective cohort pilot study	(1) pH 7.00 or base deficit 16 mEq/L in umbilical arterial cord plasma OR (2) history of an acute perinatal event and either no blood gas available or a pH from cord arterial serum ranging from 7.01 to 7.15 or a base deficit from 10 to 15.9 mEq/L AND (3) 10 min APGAR score <5 OR assisted ventilation at birth, AND moderate or severe encephalopathy	WBH for 72 h (NICHD)	/	Within the first 24 h	Scr assay: not mentioned cooling blanket (Blanketrol II)
La Haye-Caty et al. 2020 [41]	202	Retrospective review	(1) History of an acute perinatal event, cord pH ≤7.0 or base deficit ≤−16 mEq/L OR (2) evidence of neonatal distress, such as an APGAR score ≤5 at 10 min, postnatal blood gas pH obtained within the first hour of life ≤7.0 or base deficit ≤−16 mEq/L OR (3) a continued need for ventilation initiated at birth and for at least 10 min AND (4) moderate to severe neonatal encephalopathy	WBH for 72 h (NICHD)	/	Scr on admission; highest Scr during hospitalization; difference between both	Scr assay: not mentioned
Gupta et al. 2016 [42]	106	Retrospective review	(1) Metabolic acidosis OR (2) need for prolonged resuscitation AND moderate to severe encephalopathy	WBH for 72 h (NICHD)	/		Scr assay: Jaffe, Siemens Dimension RXL Chemistry Analyzer
Tanigasalam et al. 2016 [43]	60	Randomized controlled trial	(1) pH ≤7 or base deficit ≥12 mEq in cord blood (2) AND 2 of the following: APGAR 10 min ≤5, fetal distress, assisted ventilation for at least 10 min after birth, evidence of any organ dysfunction AND encephalopathy.	WBH for 72 h (*n* = 60)	Standard treatment (*n* = 60)	6 h, 36 h, 72 h	Scr assay: not mentioned. Pre-cooled gel packs (±4) to keep core temperature between 33–34 °C (chest, abdomen, back, head, axilla). Continuous rectal and skin temperature monitoring. Every 15 min for the first four h, every 2 h for the next 68 h. After cooling, gel packs were removed and radiant warmer was set at 0.5 °C/h to reach the target temperature of 36.5 °C in the next 6 h.
Selewski et al. 2013 [44]	53	Retrospective review	Cool Cap trial	WBH (*n* = 53) (NICHD)	SHC (*n* = 43)	6 h, 24 h, 48 h, 72 h, d5, d7, d10 (as clinically indicated)	Scr assay: not mentioned
Frymoyer et al. 2013 [45]	61	Retrospective chart review	One or more of the following: APGAR score <5 (at 10 min of life), history of prolonged resuscitation at birth, presence of severe acidosis defined as a cord pH or any arterial or venous pH <7 within 60 min of birth, or a base deficit >−12 from cord blood or any arterial blood gas within 60 min of life AND moderate–severe encephalopathy	WBH for 72 h with gentamicin at Q36 (*n* = 27) OR WBH for 72 h with gentamicin at Q24 (*n* = 34)	/	d2	Scr assay: not mentioned. Blanket cooling device (Cincinnati Subzero Blanketrol III). Rectal monitoring keeping core temperature at 33.5 °C.
Chock et al. 2018 [46]	38	Retrospective review	NICHD criteria	WBH for 72 h (NICHD)	/	d1, d2, d3, d4	Scr assay: enzymatic (isotope dilution mass spectrometry traceable, IDMS). Cincinnati Sub-Zero Hyper- Hypothermia Blanketrol system (Cincinnati Sub-Zero, Cincinnati, Ohio).
Oncel et al. 2016 [47]	41	Prospective nested case-control study	TOBY study criteria AND Sarnat stage II or III	WBH (*n* = 41)	Healthy controls (*n* = 20)	24 h, 48 h, 72 h results: d1, d4	Scr assay: not mentioned Cooling at rectal temperature of 33–34 °C with Tecotherm TS med 200 n (Inspiration Healthcare Ltd., Leicester, UK). After cooling, rewarming with 0.5 °C, reaching normal body temperature within 8 h.
Battin et al. 2003 [48]	13	Randomized controlled trial	(1) Gestational age >37 w OR (2) 5 min APGAR <6 OR cord/first pH <7.1 AND (3) encephalopathy consisting of lethargy/stupor, hypotonia, and abnormal reflexes including an absent or weak suck	SHC (rectal T 34.5–35 °C, WBH) (*n* = 13)	SHC + normothermic (*n* = 13)		Scr assay: not mentioned Cooling cap: Silclear tubing (Degania Silicone Ltd., Degania Bet, Israel) and a commercially made device (Olympic Medical, Seattle, WA). Initial water temperature: 10 °C. Continuous monitoring of rectal, fontanelle and nasopharyngeal temperature and keeping rectal temperature at 34.5 °C ± 0.5 °C–35 °C ± 0.5 °C or normothermic. After cooling, rewarming at 0.5 °C/h until temperature reached the normal range.

**Table 3 children-08-00475-t003:** Patient characteristics (i.e., gestational age, birth weight, APGAR score at 5 and 10 min, %female) as reported in the retained studies of this meta-analysis (IQR: interquartile range) [36,37,38,39,40,41,42,43,44,45,46,47,48].

	Lee et al. [36]	Sarkar. et al. 2009 [37]	Sarkar el al. 2014 [38]	Roka et al. [39]	Chalak et al. [40]	La Haye-Cathy et al. [41]	Gupta et al. [42]	Tangi-salam et al. [43]	Selewski et al. [44]	Frymoyer et al. [45]	Chock et al. [46]	Oncel et al. [47]	Battin et al. [48]
**mean gestational age (w)**	38 6/7 ± 1 5/7	38.5 ± 1.7	/	/	39 ± 2	39.15 ± 1.6	38.7	39.5 ± 1.3	39 ± 1.6	39.7 ± 1.6	38.6 ± 2	38.7 ± 1.6	40.1 ± 1.6
**mean birth weight (g)**	3161 ± 869	3112 ± 755	/	/	3156 ± 624	3375 ± 626	3305	2690 ± 340	3313 ± 618	3340 ± 600	3258 ± 653	3264 ± 509	3634 ± 598
**median Apgar 5 min (IQR)**	4 (2–5)	68%: 0–3, 20%: 4–5, 12%: ≥6	/	/	6(5–7)	/	/	/	(Median, SD) 3 ± 2	3.4 ± 2	4(0–9)	30% < 5	4.5 (0–7)
**median Apgar 10 min (IQR)**	5 (3–7)	33%: 0–3, 45%: 4–5, 22%: ≥6	/	/	/	/	/	5 (3–6)	/	5 ± 2	/	/	/
**female (%)**	41%	36%	/	/	30%	44%	41%	/	43%	/	44%	77%	/

**Table 4 children-08-00475-t004:** Serum creatinine (Scr) values as reported in the retained studies of this meta-analysis (HT: hypothermia; RW: rewarming; NT: normothermia; SD: standard deviation, IQR: interquartile range) [36,37,38,39,40,41,42,43,44,45,46,47,48].

Study	Outcome Measure	Median Scr (Range) 0–72 h	Maximal Scr	Scr,6 h	Scr, 24 h	Scr, 36 h	Scr, 48 h	Scr, 72 h	Scr, 96 h	Scr, 5 d	Scr, 7 d
Lee et al. 2017 [36]	Scr during HT–NT–RW, (mean, SD)		0.9 (0.6)								
Sarkar et al. 2009 [37]	Scr, during HT (median, IQR)		1 (0.8, 1.4)	1 (0.8, 1.3)							
Sarkar et al. 2014 [38]	Scr, during HT + RW (mean, SEM)			0.9 ± 0.1	0.9 ± 0.2		0.9 ± 0.3	0.8 ± 0.4		0.8 ± 0.4	
Róka et al. 2007 [39]	Scr during HT (median, IQR)			1.04 (0.87–1.15)	0.8 (0.73–0.99)		0.65 (0.5–0.74)	0.57 (0.47–0.69)			
Chalak et al. 2014 [40]	Scr (median, IQR)				1 (0.8–1.5)						
La Haye-Caty, et al. 2019 [41]	Scr (mean, SD, or median, IQR)		1.02 ± 0.3	0.95 ± 0.29	highest 0.95 (0.79–1.13)		highest 0.66 (0.5–0.92)	highest 0.48 (0.35–0.66)	highest 0.45 (0.36–0.66)		
Gupta et al. 2016 [42]	Scr (median)				1.09			0.66		0.65	0.51
Tanigasalam et al. 2016 [43]	Scr (mean, SD)			1.15 ± 0.31		1.13 ± 0.33		1.195 ± 0.55			
Selewski et al. 2013 [44]	Scr (mean, SD)			0.97 ± 0.26	1 ± 0.42		0.91 ± 0.57	0.85 ± 0.65		0.79 ± 0.75	0.74 ± 0.88 discharge: 0.49 ± 0.49
Frymoyer et al. 2013 [45]	Scr, range or mean, SD	range (d2): 0.5–1.5					1 ± 0.2				
Chock et al. 2018 [46]	Scr (mean, SD)				1.2 (0.23)		0.9 (0.34)	0.7 (0.37)	0.6 (0.40)		
Oncel et al. 2016 [47]	Scr (mean, SD)				1.02 ± 0.37			0.83 ± 0.5			
Battin et al. 2003 [48]	Scr (mean, SD)	1.6 ± 0.76									

## Data Availability

The analysis and the data presented in this study are available upon reasonable request from the corresponding author.

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
