# Peer review of "Creatinine Trends and Patterns in Neonates Undergoing Whole Body Hypothermia: A Systematic Review"

_children, 2021, doi:10.3390/children8060475_

Round 1

Reviewer 1 Report

This systemic review attempted to educate the readers regarding serum creatinine trends in HIE patient undergoing therapeutic hypothermia. Literature search was done extensively, but in my opinion the data collected/available was limited.  HIE is multiorgan condition, it is not surprising that the babies who have moderate to severe hypoxic ischemic encephalopathy have kidney injury to some extent, so their creatinine can be high before TH started or rise within first 24hrs and stay high for some time.  There is no clear-cut correlation between GFR and Serum Creatinine in neonates and the fact that initial creatinine value can be reflective of maternal creatinine makes it even more difficult to base decision on creatinine value only, but if it is persistently high, along with other abnormalities like oliguria, hyperkalemia etc. It will warrant further investigation and management. In this systemic review maternal characteristics, and Classification of HIE is not documented and there is so much variability in Creatinine value between studies, this review adds limited value to current knowledge. I think it will be beneficial to review the included articles again/reach out to authors to have more complete data and do more extensive statistical analysis.

Authors did extensive database search and it takes major part of manuscript but the statistics are very simple and end point is not very clear like how this paper would contribute to existing knowledge.

Need to improve on stats and language of the manuscript

Author Response

Reviewer 1

This systemic review attempted to educate the readers regarding serum creatinine trends in HIE patient undergoing therapeutic hypothermia. Literature search was done extensively, but in my opinion the data collected/available was limited. 

We thank the reviewer for the overall positive assessment of the systematic review conducted, as this systematic review has indeed been conducted based on an extensive (different sources) and valid systematic review (two pairs of eye concept, discussion).

We agree that the data and the final results that could be collected do not have the granularity we hoped for or anticipated. Even following our repeated attempts to get into contact with the corresponding authors failed to a certain extent, as also mentioned in the discussion part of the paper as this is somewhat concerning in a ‘data sharing concept’. However, we cannot be blamed for the final results and we read this comment as a support for the methods, while the outcome is more a ‘priority’ issue. At least, we have been able to plot (figures of the paper) mean/median trends over postnatal age, while the data (cfr comments reviewer 2) indeed were not sufficiently informative to add standardized indicators of uncertainty or variability (SD, IQR, or range) to these figures. Consequently, we remained very cautious on our conclusions and descriptive, and have provided suggestions on next steps to be taken to better catch the pattern and its variability.

 HIE is multiorgan condition, it is not surprising that the babies who have moderate to severe hypoxic ischemic encephalopathy have kidney injury to some extent, so their creatinine can be high before TH started or rise within first 24hrs and stay high for some time. 

 We obviously agree on this, as we have recently summarized the available data on Acute Kidney Injury (AKI) incidence in asphyxia cases + whole body hypothermia (Allegaert et al, Front Pediatr 2021, REF 27), and as it has recently been reported that the incidence of AKI is reduced in WBH cases when compared to moderate to severe HIE cases not undergoing WBH (van Wincoop et al, PLoS One 2021, REF 21). AKI occurs in 39-42 % of these cases (Kirkley et al, 2019, REF 14).

However, AKI is a dichotomous outcome variable, currently based on the pRIFLE criteria or the neonatal AKI definition and necessitates repeated Screa measurements, while we aimed to quantify the mean/median trends and differences in WBH cases compared to a reference dataset. In our opinion, the novelty is in the pooled description of the pattern in Screa in WBH cases compared to a reference dataset. This has its relevance, as this type of information can be used to estimate changes in renal clearance to feed PK and PBPK prediction models. Other results are the poor reporting on creatinine assays, and the uncertainties in the second part of the first week of life. This information can be useful to guide future pooled efforts.

 There is no clear-cut correlation between GFR and Serum Creatinine in neonates and the fact that initial creatinine value can be reflective of maternal creatinine makes it even more difficult to base decision on creatinine value only, but if it is persistently high, along with other abnormalities like oliguria, hyperkalemia etc. It will warrant further investigation and management.

 We agree, but there are some GFR formulae (orginal Schwatz, adapted Schwatz) that can be used. The issue on the reflection of maternal creatinine is well taken, but has also major limitations, as creatinine is only very rarely collected at delivery in the mothers, and also fails to be a ‘perfect’ reflection of the creatinine in the newborn at birth. We have very recently reported on this aspect (Allegaert et al, Neonatology 2020: creatinine at birth correlates with gestational age and birth weight: another factor of the imbroglio in early neonatal life), and hereby confirmed the recently reported findings of Go et al. Creatinine at birth is somewhat higher in the most mature neonates compared to the preterm cases with a positive correlation with birth weight or gestational age. A similar pattern has been described by Go et al. PLoS One 2018, but these authors also prospectively co-collected maternal creatinine values (figures displayed). This correlation of neonatal creatinine with GA or weight is in contrast to stable maternal creatinine values in this gestational time interval. We have added this information and the Go reference to the revised version of the paper.

 In this systemic review maternal characteristics, and Classification of HIE is not documented and there is so much variability in Creatinine value between studies, this review adds limited value to current knowledge.

The search and analysis was restricted to moderate and severe asphyxia cases all undergoing whole body hypothermia, so classification of HIE was included in the analysis, as data on serum creatinine values in moderate versus severe cases subpopulations were almost never reported in the source documents.

Furthermore, we have extracted data on the serum creatinine assays (Jaffe or enzymatic, IDMS traceable) as reported, but also this methodological aspect is rather poorly reported in the retained papers (cfr table 2, last colom). This has been further stressed in the discussion part of the paper, as this matters and affects absolute values, likely better to handle if assays used in the clinical setting all will be IDMS traceable. This is in our assessment, a valuable observation of the current analysis

Maternal characteristics were neither retained (those were not included in the ‘key search indicators’ of the study protocol), but are almost never reported, and maternal creatinine (cfr response to the former question) were never reported, so that we unable to provide information on this aspect.

I think it will be beneficial to review the included articles again/reach out to authors to have more complete data and do more extensive statistical analysis. Authors did extensive database search and it takes major part of manuscript but the statistics are very simple and end point is not very clear like how this paper would contribute to existing knowledge. Need to improve on stats and language of the manuscript

As mentioned earlier in this rebuttal, as well as in the paper, a reach out to the corresponding authors of the papers retained has been done for this paper, but was only of very limited success (one author replied, as mentioned in the paper). As a consequence, we felt that it would statistically be ‘overreaching’ and inaccurate to merge the available mean/median values to one single value, but we ‘simply’ plotted the trends as described in the individual cohorts to create a ‘snapshot’, as subsequently applied ‘visual inspection’ and interpretation, similar to what is commonly done to assess the quality of popPK plots. Such visual inspection is however also accepted in the development and reporting of such popPK plots, and is explicitly mentioned in guidelines of authorities, like FDA or EMA (like https://www.ema.europa.eu/en/documents/scientific-guideline/guideline-reporting-results-population-pharmacokinetic-analyses_en.pdf). We hereby are fully aware that this approach has its limitations as also discussed and further extended in the revised version, but felt that these limits related to the currently accessible information.

Reviewer 2 Report

Understanding the natural history of Scr in children undergoing WBH is potentially valuable. However, this analysis appears haphazard in data inclusion and did not perform fully quantitative analysis. Visual analysis without formal statistics is not sufficient to draw conclusions reached regarding Scr trends.

Regarding study inclusion:

-It is concerning that the search strategy missed a relevant paper that was ultimately included-- perhaps suggesting an inaccurate search strategy

-As most studies were excluded, it is important to understand how many were excluded for each reason

-Restricting analysis to free-access manuscripts is not appropriate

-Given emphasis on WBH, it is not clear why head cooling studies were included

Data analysis:

-Visual inspection is inadequate to reach solid conclusions (particularly as not all studies in Figure 3 even clearly downtrend over displayed data points)

-Some attempt to pool results across studies is needed. Ideally this would use multivariate analysis to control for factors across studies 

-Figure 3 (upon which analysis/conclusions seem to be reached) does not weight based on sample size or include error bars to indicate degree of uncertainty. As such it is difficult to identify visually whether, for example, as claimed, whether there is a significant decline from day 2 to day 4

Author Response

Reviewer 2

Understanding the natural history of Scr in children undergoing WBH is potentially valuable. However, this analysis appears haphazard in data inclusion and did not perform fully quantitative analysis. Visual analysis without formal statistics is not sufficient to draw conclusions reached regarding Scr trends. 

The natural history of Scr in children undergoing WBH is in our opinion valuable to develop and explore popPK and PBPK models, besides the links between AKI and neurocognitive outcome, and mortality (mentioned in the paper, REF 15, REF 17). It is therefore useful to describe and compare these trends as AKI is a dichotomous variable that commonly necessitates repeated Screa measurements over time. Finally, the pattern matters to take next steps, as this should guide us in the study design (like eg focus on the first 8-10 days of life, focus on information on the assay used, focus on outcome aspect, the absence of Cystatin C values at present). This has been somewhat further extended in the discussion part of the paper.

We agree that a fully quantitative analysis was not conducted, as this was not feasible and scientifically not robust in our opinion based on the data as accessible. We have discussed this limitations higher and lower in the rebuttal, but felt that we could not simply ‘merge’ mean or median value to generate another ‘pooled’ mean or median value.

Regarding study inclusion:

-It is concerning that the search strategy missed a relevant paper that was ultimately included-- perhaps suggesting an inaccurate search strategy

We respectfully disagree, as the relevant paper (Mok et al, Sc Report 2020, published 24 Sept 2020) has been published after the search strategy was finalized. This is the very reason that we have reported on this additional paper in the discussion part of the paper, and we felt that an additional figure (figure 4 in addition to figure 3) was the best approach, as the information was available at submission, not yet at search conductance (01.05.2020, mentioned in the paper). To a certain extent, this is somehow an ‘external’ validation’ of our initial visual plot. We have further added this information to the revised version of the paper.

-As most studies were excluded, it is important to understand how many were excluded for each reason

The 135 excluded cases are mentioned in Figure 2, but there is quite some overlap in exclusion criteria (full text, Scr values and/or no hypothermia), so that we suggest to keep this Figure 1 as currently provided.

-Restricting analysis to free-access manuscripts is not appropriate

We agree, but this is not how the study has been conducted. Full text retrieval was based on the title and abstract. ‘When the full text was not found, we tried to contact the corresponding author to request a full text version’. This is for sure not restricted to open access, but also included access to local libraries (KU Leuven, or Erasmus MC), and by the local KU Leuven library to an (inter)national network of libriaries, LIMO) to provide a scanned version. If all failed, we used e-mail to the corresponding author (if retrieved, not so common in abstract). We have rephrase the methods section.

-Given emphasis on WBH, it is not clear why head cooling studies were included

‘Selective’ head cooling was not included, but some studies used these criteria to initiate cooling, while other studies used the NICHD criteria, so that we felt we have to provide this information. Furthermore, for the  ‘Sarkar et al, 2009 and the Battin et al, 2003 cohorts: these were tRCT on selective head cooling compared to WBH, but only the data in WBH cases were included in the current analysis.  

Data analysis:

-Visual inspection is inadequate to reach solid conclusions (particularly as not all studies in Figure 3 even clearly downtrend over displayed data points)

The ‘visual inspection’ approach taken, and interpretation obviously has its limitations, but is similar to what is commonly done to assess the quality of popPK plots. Such visual inspection is however also accepted in the development and reporting of such popPK plots, and is explicitly mentioned in guidelines of authorities, like FDA or EMA (like https://www.ema.europa.eu/en/documents/scientific-guideline/guideline-reporting-results-population-pharmacokinetic-analyses_en.pdf). We hereby are fully aware that this approach has its limitations as also discussed and further extended in the revised version, but felt that these limits related to the currently accessible information. We have further stressed this is in the discussion section of the paper (limitations section).

-Some attempt to pool results across studies is needed. Ideally this would use multivariate analysis to control for factors across studies 

We agree, but we have tried such an attempt by contacted some of the corresponding authors to get more detailed information, and even this first attempt failed. We fully agree that an meta-analysis based on individual data is an obvious next step. In our opinion, the value of this report is that this can inform us on the relevant data and the study design (cfr higher). This has been added in the discussion part of the paper.

-Figure 3 (upon which analysis/conclusions seem to be reached) does not weight based on sample size or include error bars to indicate degree of uncertainty. As such it is difficult to identify visually whether, for example, as claimed, whether there is a significant decline from day 2 to day 4

We agree, and therefore we were already very cautious on ‘firm’ conclusions, and have further elaborate on the limitations in the revised version of the paper.

Round 2

Reviewer 1 Report

The authors have tried to address reviewer's feedback and revised the paper. With the current data available, I do not think the authors can add more to manuscript at this time and agree that attention to S creatinine trends and renal function is warranted in neonates affected by HIE

Author Response

to the best of our knowledge, there were no additional comments from reviewer 1 and the reviewer accepted the former version already. 

Reviewer 2 Report

Thank you for your detailed responses. I recognize the difficulties posed by incomplete data and lack of individual-level data. Some methodological concerns I had appear to be reasonably-handled, though text should reflect analysis steps more clearly. I continue to argue that data presentation must be improved (particularly Figure 3 as all conclusions are drawn from data presented there) and that results description/conclusions be softened to better reflect what has and has not been established. With more precise discussion of uncertainty, I now feel that this article can make some addition to the literature.

Methods:

-I agree with the decision to include only participants who underwent WBH (whether or not they also received head cooling). This needs to be more clear in the text (e.g. inclusion criteria in Line 142 makes no mention of hypothermia type; Lines 150-151 as stated is not clear; Lines 196-197 are also not clearly stated).

-Inclusion criterion 3 (Line 143) requiring normothermic control groups seems to contradict the control column of Table 2, in which most included studies appeared not to have a normothermic control group. Please confirm that stated inclusion criteria are accurate

-I understand that articles may have been excluded for multiple reasons but still feel that is is important to understand how many were excluded due to full text not being retrieved (as in PRISMA guidelines Figure 1 https://www.bmj.com/content/372/bmj.n71, separating screening exclusions from records not retrieved). If not reported in Figure 1, this information should be included in the text.

Results:

-I recognize that visual inspection has potential value, but Figure 3 must be clarified to best permit visual analysis.

-Vertical error bars are needed to reflect uncertainty of group values

-Small studies are given equal visual weight to large studies. I recommend reflecting sample size/some measure of study quality via, for example, line weight or point weight

-Does the distinction between solid vs. dotted lines indicate a difference in study type? If not, are there any studies with differences in methodology that should be visually distinguished?

-Regarding description of results:

-Via visual inspection, I agree that it is clear that Scr is greater than reference ranges at age <6h. By 24h or 48h however, this is less clear, and I would not be able to conclude that Scr is clearly higher than reference ranges at those times-- let alone that elevations are clinically relevant. If making a claim of clinical relevance, a priori criteria defining a threshold of clinically-relevant elevations would be helpful.

-I do not find the claim of "different slope" (Line 260) substantiated. To even support a finding of different slopes statistically, this would require analyses of individual trajectories, which are not available in this analysis

Discussion:

-Line 280: I would not describe this approach as "comprehensive"

-Line 294 and throughout: I would not use the word "significantly" in the absence of statistics

-Lines 299-302: I am not sure that the problem is one of sample size or length of follow-up so much as lack of validation against more direct measures of, e.g., GFR

-Line 324: Again I have not yet been convinced that elevations are clinically relevant

-Consideration of Figure 4 and corresponding text belongs in Results rather than discussion. I remain unconvinced that this represents a "blunted decrease" rather than persistence of borderline elevations-- again, error bars would be helpful for interpretation.

Conclusions:

-Line 371: Again please do not describe increases as significant

Author Response

Thank you for your detailed responses. I recognize the difficulties posed by incomplete data and lack of individual-level data. Some methodological concerns I had appear to be reasonably-handled, though text should reflect analysis steps more clearly. I continue to argue that data presentation must be improved (particularly Figure 3 as all conclusions are drawn from data presented there) and that results description/conclusions be softened to better reflect what has and has not been established. With more precise discussion of uncertainty, I now feel that this article can make some addition to the literature.

Methods:

-I agree with the decision to include only participants who underwent WBH (whether or not they also received head cooling). This needs to be more clear in the text (e.g. inclusion criteria in Line 142 makes no mention of hypothermia type; Lines 150-151 as stated is not clear; Lines 196-197 are also not clearly stated).

We have further stressed this in this section, so that it is clear that any whole body hypothermia, irrespective of additional interventions provided (like in the RCT on WBH, with or without selective head cooling). To do so, we have added ‘whole body therapeutic hypothermia (irrespective of add on interventions like additional selective head cooling’ in the former L 142, 150-152 and former line 196-197).

-Inclusion criterion 3 (Line 143) requiring normothermic control groups seems to contradict the control column of Table 2, in which most included studies appeared not to have a normothermic control group. Please confirm that stated inclusion criteria are accurate

We have adapted this. The control group is the reference Scre dataset in (near)term cases. This has been further highlighted in the revised version of the abstract and in the methods section.

-I understand that articles may have been excluded for multiple reasons but still feel that it is important to understand how many were excluded due to full text not being retrieved (as in PRISMA guidelines Figure 1 https://www.bmj.com/content/372/bmj.n71, separating screening exclusions from records not retrieved). If not reported in Figure 1, this information should be included in the text.

The major reasons of exclusion at the stage of eligibility were mainly being not relevant to the topic like post mortem pathology findings, animal experimental studies or language, with only failure to retrieve one conference abstract. There were 4 conference abstracts, but we could only retrieve 3 of these, but none of them was retained in the subsequent screening to end up in the qualitative synthesis.

Results:

-I recognize that visual inspection has potential value, but Figure 3 must be clarified to best permit visual analysis.

-Vertical error bars are needed to reflect uncertainty of group values:

We have added ‘error bars’ to the figure 3, but this results in overcrowded picture, while these data on variability (SD, IQR, range) provided in Table 4. As this information has not been provided in the source document (Mok et al., see also lower: we have also repeatedly contacted the corresponding author, but he decided not to share any information, neither SD, nor raw data) plotted in Figure 4, we are not able to add this information to Figure 4. We therefore felt that adapting the legend with an explicit referral to Table 4. We hope that this is a reasonable, and better option (we are not able to upload the figure on the comments to the reviewers section, so we have added the rebuttal as an additional document).

Figure 3, with the error bars added looks like this.

We therefore assume that the reviewer agrees that this does not provide add on benefit to the former figure 3, so that we suggest to restrict the changes to the adding the numbers to reflect the size of the different cohorts (cf comment below).

-Small studies are given equal visual weight to large studies. I recommend reflecting sample size/some measure of study quality via, for example, line weight or point weight.

We already already reflected on how to present such data, as the combined vertical error, the different colors and the add another layer of information on study size is not feasible. We therefore decided to add this information in the table 2 (second colomn), and have further added the number of cases in the legend in the figure, with additional referral to the relevant tables (table 4 for the source of variability in Scr values, table 2 for the size of the different datasets).

-Does the distinction between solid vs. dotted lines indicate a difference in study type? If not, are there any studies with differences in methodology that should be visually distinguished?

No, this was simply because we needed all different studies plotted with colors/plotting that are sufficiently different. The methodological aspects of the individual studies has been extensively provided in Table 2, Table 3 and Table 4 to fully provide access to the granularity of these aspects, and has been added in the legend of Figure 3.

-Regarding description of results:

-Via visual inspection, I agree that it is clear that Scr is greater than reference ranges at age <6h. By 24h or 48h however, this is less clear, and I would not be able to conclude that Scr is clearly higher than reference ranges at those times-- let alone that elevations are clinically relevant. If making a claim of clinical relevance, a priori criteria defining a threshold of clinically-relevant elevations would be helpful.

We have adapted the wording ‘significant’, or clinical relevant throughout the paper, but still Scr remains an indicator of subsequent outcome (both mortality, and morbidity) in this specific populations (among others ref 7,14). Assessing Figure 3, all mean or median observations (n=18) of the different cohorts are above the 50th centile of the reference dataset at <6h, 24 h and 48 h respectively, and none of the points in the first 48 h are  ≤50th centile. This quantitative and more objective finding has been added to the revised version of the paper, instead of the descriptive text line, but confirms the visual inspection.

-I do not find the claim of "different slope" (Line 260) substantiated. To even support a finding of different slopes statistically, this would require analyses of individual trajectories, which are not available in this analysis

This text section has been rephrased, including removal of the ‘different slope’ wording.

Discussion:

-Line 280: I would not describe this approach as "comprehensive"

removed

-Line 294 and throughout: I would not use the word "significantly" in the absence of statistics

removed

-Lines 299-302: I am not sure that the problem is one of sample size or length of follow-up so much as lack of validation against more direct measures of, e.g., GFR.

We are only aware of our own recent study on mannitol clearance, as an exogenous marker to quantify clearance, so we have added this information to the paper. We assume that the reviewer wanted to restress the limitations of Scr as indicator of GFR, and that’s correct, so that we have repeated this overhere. However, Scr as AKI has been used as an indicator and predictor of outcome (both morbidity and mortality. Merging these concept, we have add this sentence:  “Furthermore, the link between Scr and GFR as direct measure of renal function is only poorly validated in early neonatal life. However, there are data on differences in mannitol clearance as another indicator of GFR impairment, while the link with neurocognitive outcome and mortality is based on the AKI definition, based on Scr and diuresis, not GFR [7,11,14,25].”

-Line 324: Again I have not yet been convinced that elevations are clinically relevant

removed

-Consideration of Figure 4 and corresponding text belongs in Results rather than discussion. I remain unconvinced that this represents a "blunted decrease" rather than persistence of borderline elevations-- again, error bars would be helpful for interpretation.

Based on the information provided in the paper, we only have access to the mean (Figure 2 of the Mok paper) data in AKI and non-AKI data, and also after repeated attempts, the corresponding author preferred not to share this information (or raw data), despite the specific mentioning to be willing to share data upon reasonable request, so that we cannot add additional information in Figure 4 of this paper. We still wanted to make it clear that this Mok paper was no part of the initial search, so prefer to keep it out of the results section. However, to align this to the request of the reviewer, we have added a section in the methods section on the additional non-systematic search after finalization of the systematic search, but before submission as an attempt to add to the available information as ‘external validation’ type of approach. By this construct, we could subsequent add this Figure 4 to the results section and have left the reflections on this in the discussion section.

Conclusions:

-Line 371: Again please do not describe increases as significant

Removed
